# A diploid assembly-based benchmark for variants in the major histocompatibility complex

Chen-Shan Chin[1], Justin Wagner[2], Qiandong Zeng[3], Erik Garrison[4], Shilpa Garg[5], Arkarachai Fungtammasan[1], Mikko Rautiainen[6,7,8], Sergey Aganezov [9], Melanie Kirsche [9], Samantha Zarate[9], Michael C. Schatz[9,10], Chunlin Xiao [11], William J. Rowell [12], Charles Markello[4], Jesse Farek[13], Fritz J. Sedlazeck [13], Vikas Bansal[14], Byunggil Yoo [15], Neil Miller [15], Xin Zhou[16], Andrew Carroll[17], Alvaro Martinez Barrio [18], Marc Salit[19], Tobias Marschall [20], Alexander T. Dilthey [21] & Justin M. Zook [2]✉

Most human genomes are characterized by aligning individual reads to the reference genome, but accurate long reads and linked reads now enable us to construct accurate, phased de novo assemblies. We focus on a medically important, highly variable, 5 million base-pair (bp) region where diploid assembly is particularly useful - the Major Histocompatibility Complex (MHC). Here, we develop a human genome benchmark derived from a diploid assembly for the openly-consented Genome in a Bottle sample HG002. We assemble a single contig for each haplotype, align them to the reference, call phased small and structural variants, and define a small variant benchmark for the MHC, covering 94% of the MHC and 22368 variants smaller than 50 bp, 49% more variants than a mapping-based benchmark. This benchmark reliably identifies errors in mapping-based callsets, and enables performance assessment in regions with much denser, complex variation than regions covered by previous benchmarks.

[1] DNAnexus, Inc, 1975 W El Camino Real, Suite 204, Mountain View, CA 94040, USA. [2] Material Measurement Laboratory, National Institute of Standards and Technology, 100 Bureau Dr, MS8312, Gaithersburg, MD 20899, USA. [3] Laboratory Corporation of America Holdings, 3400 Computer Drive, Westborough, MA 01581, USA. [4] University of California, Santa Cruz, 1156 High St, Santa Cruz, CA 95064, USA. [5] Department of Genetics, Harvard Medical School, Boston, MA, USA. [6] Center for Bioinformatics, Saarland University, Saarland Informatics Campus E2.1, 66123 Saarbrücken, Germany. [7] Max Planck Institute for Informatics, Saarland Informatics Campus E1.4, 66123 Saarbrücken, Germany. [8] Saarland Graduate School for Computer Science, Saarland Informatics Campus E1.3, 66123 Saarbrücken, Germany. [9] Department of Computer Science, Johns Hopkins University, Baltimore, MD 21218, USA. [10] Simons Center for Quantitative Biology, Cold Spring Harbor Laboratory, Cold Spring Harbor, New York, NY 11724, USA. [11] National Center for Biotechnology Information, National Library of Medicine, National Institutes of Health, 8600 Rockville Pike, Bethesda, MD 20894, USA. [12] Pacific Biosciences, Menlo Park, CA 94025, USA. [13] Human Genome Sequencing Center, Baylor College of Medicine, One Baylor Plaza, Houston, TX 77030, USA. [14] Department of Pediatrics, University of California San Diego, La Jolla, CA 92093, USA. [15] Genomic Medicine Center, Children's Mercy Kansas City, Kansas City, MO 64108, USA. [16] Department of Computer Science, Stanford University, Stanford, CA 94305, USA. [17] Google Inc, 1600 Amphitheatre Pkwy, Mountain View, CA 94043, USA. [18] 10x Genomics, Pleasanton, CA 94588, USA. [19] Joint Initiative for Metrology in Biology, Stanford, CA 94305, USA. [20] Institute of Medical Biometry and Bioinformatics, Medical Faculty, Heinrich Heine University Düsseldorf, 40225 Düsseldorf, Germany. [21] Institute of Medical Microbiology and Hospital Hygiene, Heinrich Heine University Düsseldorf, 40225 Düsseldorf, Germany. ✉email: jzook@nist.gov

The Genome in a Bottle Consortium (GIAB) and Illumina Platinum Genomes have developed benchmarks for selected easier small variants[1–3] and structural variants[4], but even the most recently published benchmarks only increased genome coverage from <80% to between 80 and 90% of the human reference genome. The short reads used to develop the small variant benchmarks cannot be uniquely mapped to many repetitive regions of the genome, such as segmental duplications, tandem repeats, and mobile elements. This includes the very challenging but medically important ~5 million base-pair (bp) region in the human genome called the Major Histocompatibility Complex (MHC). The MHC contains a set of human leukocyte antigen (HLA) genes that play crucial roles in autoimmunity and response to infection, including adaptive and innate immunity[5]. It is exceptionally variable between individuals and very challenging to characterize with conventional methods since short reads are too different from the reference to map correctly. De novo assembly of reads can help characterize the sequence in highly divergent regions like the MHC. However, with the exception of a synthetic diploid benchmark[6], previous benchmarks primarily relied on mapping-based methods that excluded a large fraction of the sequence.

Mapping to a reference was previously necessary because only very recently have de novo assemblies been able to represent both haplotypes without suffering from small indel errors due to the error-prone long reads. While human diploid assembly is currently making great strides, including fully resolving the MHC region in two haplotigs in two previous whole-genome assemblies[7,8], these assemblies still had a substantial error rates for small variants of at least 10% due to their reliance on error-prone long reads, and the individual long and ultralong read assembly incompletely resolved haplotypes. A linked-read assembly also resolved much of the MHC for both haplotypes, but it was fragmented and had similar overall error rates for small variants[9].

Short reads have been used to assemble much of the MHC, but the assembly was highly fragmented even for haploid cell lines[10]. The current state-of-the-art methods use next-generation sequencing (NGS) to sequence a subset of exons in several HLA genes and result in a high-resolution HLA type that specifies the sequence in these exons[10]. Recently, a method to characterize the HLA at even higher resolution was shown to have the potential to improve outcomes in transplantation in a retrospective study[11]. Other genes in the MHC are also important for transplantation[12], HIV infection[13], immuno-oncology[14], and many other diseases[15]. Building on the GIAB effort to develop benchmarks (or truth sets) for the genome, this work was motivated in part by the need to benchmark the MHC. This benchmark will enable developers to optimize and demonstrate the performance of methods characterizing the MHC at increasing resolution[16].

In this work, we develop a local de novo assembly method using whole-genome sequencing (WGS) data from highly accurate long reads that are partitioned into the two haplotypes using ultralong and linked reads. We use WhatsHap[17] to combine long-range (>100 kb) phasing information from linked reads (barcoded short reads from long DNA molecules)[9], and ultralong nanopore sequencing reads. This phasing information is then used to separate highly accurate circular consensus long reads[18] into the two haplotypes for diploid assembly. We then use this diploid assembly approach to establish benchmark variant calls in a region where mapping-based methods have limitations. Specifically, we assemble both haplotypes of the MHC and use dipcall[6] to generate benchmark variants and regions in the openly consented[19] Personal Genome Project/Genome in a Bottle sample HG002 (NIST Reference Material 8391)[20]. Since most of the

MHC alternate loci in GRCh38 and other MHC sequences are not fully continuous assemblies, our assembled haplotigs represent two of only a few continuously assembled MHC haplotypes. We expect this curated benchmark set from a targeted diploid assembly will help the community improve variant calling methods and whole-genome de novo assembly methods, and form a basis for future diploid assembly-based benchmarks.

## Results

**Linked reads and long reads generate a single phase block**. We used the 10x Genomics Linked Read-based phased variant calls (84X coverage)[21], Oxford Nanopore reads (ONT, 52X total coverage and 15X coverage by reads >100 kb)[22], and PacBio Circular Consensus (HiFi, 18X coverage by 15 kb library and 16X coverage by 20 kb library) reads with predicted accuracy >99%[18] collected by GIAB (10x Genomics and PacBio) and UC Santa Cruz for establishing a high-confidence set of heterozygous marker SNVs, for phasing the corresponding variants, and generating haplotype-partitioned read sets with WhatsHap[17]. To phase this set of high-confidence SNVs, we used WhatsHap to combine phase block information from 10x Genomics' LongRanger pipeline with ONT reads[23,24], which resulted in one phased block across the entire MHC with the largest block containing all 12,441 high-confidence variants and spanning 4,949,705 bp of sequence. We compared this phasing—obtained from read data of only this one sample HG002—to a trio-based phasing using genotypes of the two parents (HG003 and HG004) and found a switch error rate of 0.23% and a Hamming error rate of 0.14%, confirming the high quality of the phasing. In particular, the low Hamming error rate shows that the phasing is correct along the whole MHC region (Fig. 1a). In our experiment, we found that we needed to utilize all three data types to achieve a single phasing block containing 12,441 confident HETs (heterozygous variants) covering the whole MHC region (Supplementary Note 1). We then used WhatsHap to use these phased variants to partition the HiFi reads into the two haplotypes (Fig. 1b). Of the 12,456 HiFi reads are recruited by alignment to GRCh37 MHC region and the MHC region from a de novo assembled contig (see below), we assigned 5413 (43.4%) reads to the first haplotype (H1, determined to be Maternal) and 5450 (43.8%) reads to the second haplotype (H2, determined to be Paternal). We were not able to determine the haplotype phase of a small number of reads (1593 reads, 12.8%), which we call untagged reads, and 15% (730,230 bp out of 4,970,557 bp) of the MHC is covered by more than 10 of these reads due to runs of homozygosity and regions highly divergent from GRCh37 and GRCh38. We used H1 and untagged reads to assemble H1, and we used H2 and untagged reads to assemble H2 (Fig. 1c).

Since an individual's MHC haplotypes can be very different from a single reference (e.g., the primary chromosomes in GRCh37 and GRCh38, which are the same), we were not able initially to generate phased contigs spanning the whole MHC region. Parts of the MHC in HG002 divergent from the GRCh37 primary reference were missing from the initial read recruitment process. We addressed this issue by generating a de novo assembly of unphased HG002 MHC contigs, and used this assembly to find an additional 335 reads (2.69% of 12,456 reads) that were not mapped to the primary GRCh37 MHC region. Without the 335 reads, the largest contig we get for an unphased assembly is 4,021,841 bp. Combining these reads with the reads mapped to GRCh37 allowed us to generate fully phased contigs (H1 and H2) covering the entire MHC region of HG002 (Fig. 1c).

**HiFi reads assembled into a single contig for each haplotype**. We used the reads that were assigned to H1 or H2 and unphased

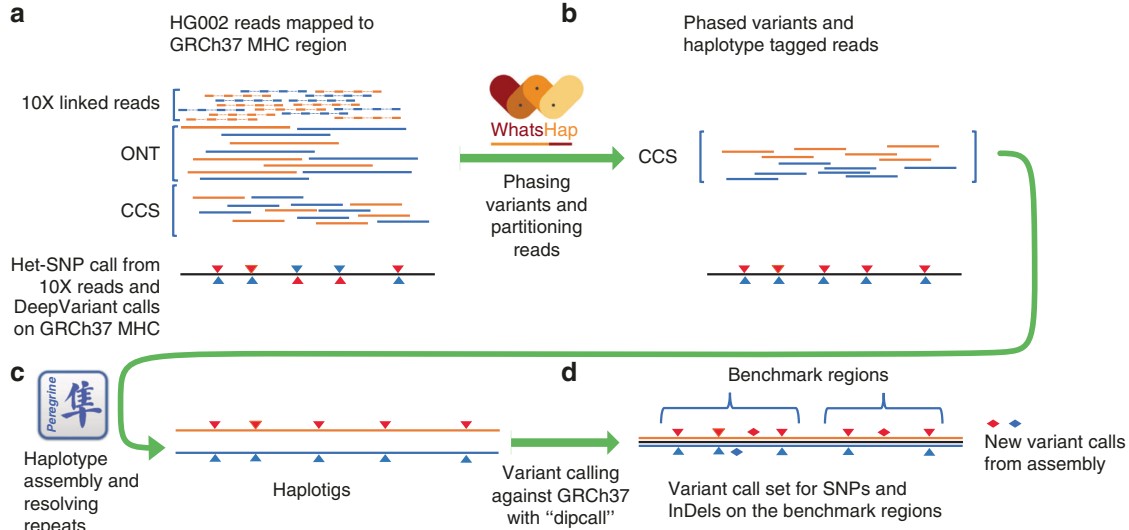

**Fig. 1 Assembling a single contig for each haplotype. a** We regenotyped DeepVariant (DV) heterozygous SNVs with WhatsHap using Oxford Nanopore Technologies (ONT) and PacBio HiFi (CCS) reads to find a confident set of SNVs with concordant genotypes from DV/CCS, WhatsHap/ONT, and WhatsHap/CCS—our Confident HETs for phasing. We selected 10x Genomics (10X) variants with phased blocks from the 10X VCF. For phasing, we used WhatsHap to combine phased blocks from 10X with ONT reads to get a single phased block across the MHC. **b** We binned PacBio HiFi reads into two haplotypes, which are denoted as orange and blue reads, using WhatsHap. **c** We performed diploid assembly using the Peregrine Assembler with the haplotype-binned HiFi reads. **d** We generated the benchmark variant callset from the assembled haplotigs using dipcall, and defined benchmark regions excluding SVs, exceptionally divergent regions, low-quality regions in the assembly, and long homopolymers.

reads as input for generating a haplotype-specific assembly. This resulted in two main haplotigs from two separate assembly processes. Unlike most existing MHC alternate loci in GRCh38, these two haplotigs cover almost the entire MHC region (Supplementary Table 1). The alignments of the haplotigs to GRCh37 are shown in Fig. 2. The alignments show a segmental duplication as well as several highly polymorphic regions, including a highly divergent region resulting in alignment gaps on both haplotypes.

There is a 30 kb segmental duplication in GRCh37 and both haplotigs containing the gene and pseudogene pairs RP, C4, CYP21, and TNX (RCCX). We use a two-step assembly approach to resolve this highly similar segmental duplication. In the first step, we allow up to 4% difference between reads when building the read overlap graph. Due to the relatively large tolerance for differences, we cannot distinguish the reads from different copies even though there are small differences between the copies. To resolve this, we introduce the second step for repeat resolution by analyzing the unique k-mers (k = 32) of each read. We classify the k-mers to be (1) erroneous k-mers and (2) haplotype/repeat-specific k-mers. With the repeat-specific k-mers, the reads from the two copies of the segmental duplication are separated before constructing the assembly graph. The alignments around the MHC Class I genes HLA-A, HLA-B, and HLA-C were very divergent, but <5% different so that the haplotigs were aligned without gaps. The only alignment gap occurred in the MHC class II genes in the 110 kb (H1), and 102 kb (H2) between HLA-DRA and HLA-DRB1, caused by the extremely high divergence that frequently occurs in this region.

We identified low confidence regions by aligning reads from each haplotype to their respective contig and finding clusters of variants (See Methods and Supplementary Table 2). There is only one such low confidence region on each haplotype, covering 10,668 bp on haplotig 1 and 14,457 bp on haplotig 2. Both of these low confidence regions are in the highly polymorphic region of the MHC and are highly divergent from GRCh37 (the white regions of Fig. 2). This region is also where additional reads needed to be recruited, so that the reads were not completely partitioned by haplotype. Still, most of the assembly in the strongly diverged region between HLA-DRA and HLA-DRB1 was supported by the reads and accurately assembled.

**Assembled contigs completely match HLA types with correct phasing.** For the genes HLA-A, HLA-B, HLA-C, HLA-DQA1, HLA-DQB1, and HLA-DRB1, we observed perfect concordance between classical HLA types and the two main haplotigs; similarly, relative to the canonical reference, the haplotigs correctly identified 157 variants across 2418 bp of sequence defining clinical HLA types (exons 2 and 3 for HLA-A, HLA-B, and HLA-C; exon 2 for HLA-DQA1, HLA-DQB1, and HLA-DRB1). Based on HLA typing data generated by a clinical laboratory on the HG002/HG003/HG004 trio previously, HLA type phasing was consistent with trio structure, and allowed us to assign H1 as the maternal and H2 as the paternal haplotype. While the main haplotigs matched the expected HLA types, we found that small extra assembled contigs contained HLA-DRB1 sequences from the opposite haplotype, presumably because of incomplete partitioning of reads in this highly complex and repetitive region. Since the main haplotigs (H1 and H2) matched the HLA types and covered the entire MHC region, and the extra contigs were short (close to the HiFi read length), we disregarded these small contigs in further analyses.

**Create a reliable small variant benchmark set from the haplotigs.** We used dipcall[6] to call variants from the main haplotigs aligned to GRCh37 (Fig. 1d). We formed small variant benchmark regions by excluding (1) the low confidence regions identified above by mapping reads to the assembly, (2) structural variants >49 bp, (3) regions with extremely dense small variants, (4) the highly divergent region including the HLA-DRB genes, and (5) perfect and imperfect homopolymers longer than 10 bp (see Methods). The vcf contains structural variants, but many of these are complex and will require new benchmarking tools

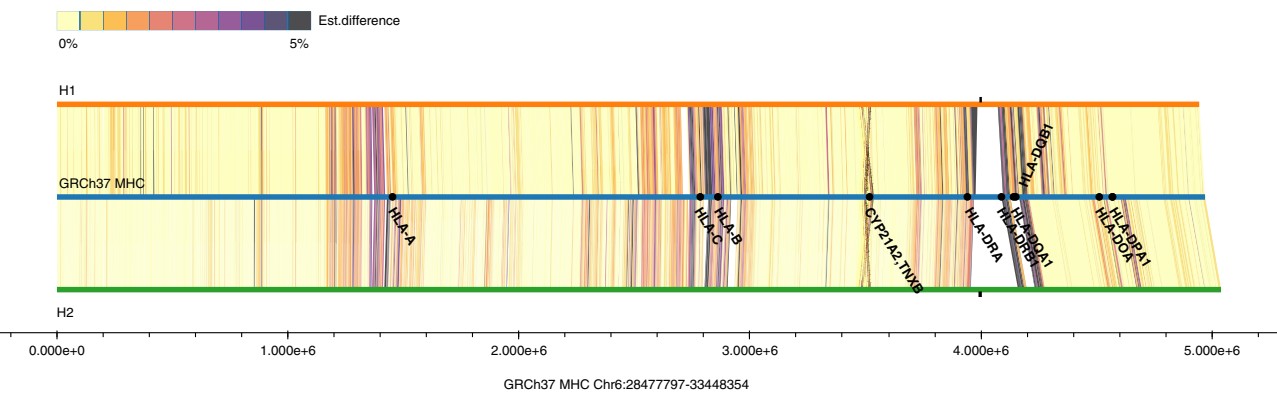

**Fig. 2 Alignments of the two main haplotigs to the primary GRCh37 MHC region.** We compute the local divergence (est. difference) of the HG002 MHC haplotigs to the MHC of GRCh37 by performing local alignment. The differences between the assembled contigs to the references are computed using sequence blocks anchored with minimers and aligned locally using an O(ND) alignment algorithm[33].

(see Supplementary Note 2), so we exclude these from the small variant benchmark regions.

To assess the accuracy of the small variant benchmarks, we compared the dipcall assembly-based variant calls to the v4.1 small variant benchmark set from GIAB. This v4.1 benchmark uses a similar variant integration approach to the previous version[1], but v4.1 adds mapping-based variant calls from the 10x Genomics and PacBio HiFi datasets used in this work to expand to regions difficult to map with short reads. Within the intersection of the v4.1 and assembly-based benchmark regions, there was one cluster of 13 putative false-positive variants in the assembly-based calls relative to v4.1 in the region 6:32597590-32597700, which were all accurately called in the assembly-based callset. In addition, there were seven 1-bp indel errors in homopolymers in the assembly benchmark due to noise in the HiFi reads, one genotyping error in a compound heterozygous indel in a tandem repeat, and one genotyping error in a compound heterozygous indel in a long homopolymer in a region of high homozygosity.

Our benchmark regions included 22,368 benchmark SNVs and indels smaller than 50 bp and covered 4.65 out of the 4.97 Mbp MHC sequence, 49% higher than the 14,999 variants included in the v4.1 alignment-based benchmark set. Our benchmark completely covered all 23 HLA genes except for (1) all of the *HLA-DRB* genes due to extreme divergence from the reference, (2) a 2 kb region covering part of one intron and exon due to extremely dense variation in *HLA-DQB1*, (3) one intronic Alu deletion, (4) 29 intronic homopolymers, (5) 4 homopolymers in UTRs, (6) 2 intronic complex variants in tandem repeats. Note that while we exclude these regions from the benchmark bed file, variants in the VCF are likely to be accurate in most of these regions except in some homopolymers, so all variants are kept in the VCF. Therefore, expert users can still compare to our variant calls or directly to our assemblies in regions like those containing the *HLA-DRB* genes.

We evaluated the utility and accuracy of our benchmark MHC small variant set by asking GIAB members to compare 11 callsets to the benchmark. To evaluate its utility across methods, the 11 callsets included mapping-based, graph-based, and assembly-based calls from Illumina, 10x Genomics, PacBio HiFi, and Oxford Nanopore. Importantly, we used *hap.py* with the *vcfeval* option to account for differences in representation of the many complex variants in the MHC[25]. To show our benchmark reliably identifies false positives (FPs) and false negatives (FNs), we manually curated 10 random FPs and 10 random FNs, half SNVs and half INDELs, and determined whether they were correct in

the benchmark and errors in the query callset (Fig. 3a, with detailed curations in Supplementary Data 1). When benchmarking against dense variant calls in divergent regions like those in our MHC benchmark, it is critical to understand that current benchmarking tools will often classify a variant as a FP when both haplotypes are not fully called correctly in the query callset (e.g., if any nearby calls are filtered), since these complex variants can be represented in many ways and current tools will not always give partial credit if some parts of the complex variant are called correctly and some called incorrectly. For example, in Fig. 4, the 5 called SNVs in the VCF are counted as FPs, even though they are consistent with the alignments of the PacBio reads, because the complex variant is not fully called (i.e., many of the variants are incorrectly filtered). For FPs like the SNVs in Fig. 4, where the variant is supported by some correctly aligned reads but the complex variant is not fully called correctly due to missing or inaccurate nearby variants, we classified these as partially correct in manual curation (Fig. 3b). We use dipcall to call the benchmark variants, because it represents complex variants as individual SNVs, insertions and deletions rather than as block substitutions. Representing the benchmark in this way makes it more likely that partial credit will be given for partially correct calls, except in the most complex cases like those in Fig. 4.

## Discussion

As one of the most polymorphic regions in the human genome, the MHC region poses many challenges for variant calling, HLA typing, and association studies. In the human reference GRCh38, there are already eight alternative MHC sequences[26]. Given that the number of observed distinct HLA alleles is still increasing, analysis of MHC haplotype structures in the whole human population will be difficult without a large number of high quality reference sequences. We anticipate that approaches using diploid assembly of long reads, like the one developed in this work, will reveal many new MHC haplotypes across the population. We show it is now possible to reconstruct highly accurate MHC haplotigs using just whole-genome shotgun sequencing. The method we describe and the resulting MHC haplotype assemblies will enrich our genomic knowledge for immunology-related diseases. In addition, variant calling from short reads will still provide significant value for accessible genotyping of variants in the MHC, and our benchmark will aid in developing and optimizing short read-based variant calling methods.

Utilizing the rich public data collection available for the GIAB sample HG002, we construct haplotigs and generate diploid

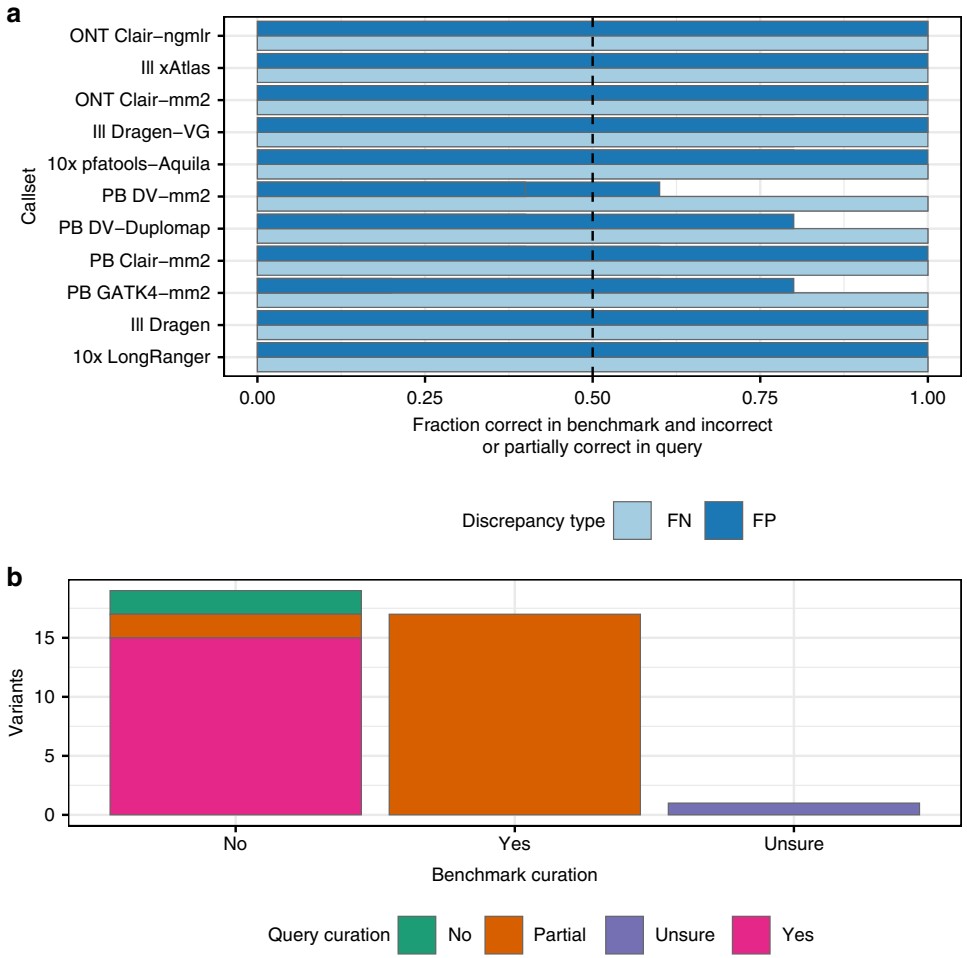

**Fig. 3 Evaluation of benchmark's ability to reliably identify FNs and FPs across technologies. a** Proportion of 10 randomly selected FPs and 10 randomly selected FNs from 11 callsets from Illumina (Ill), 10x Genomics (10x), PacBio HiFi (PB), and Oxford Nanopore (ONT) that were determined to be fully correct in the benchmark and incorrect or only partially correct in the query callset. **b** Breakdown of variants potentially incorrect in the benchmark or correct in the query, where curation of the benchmark determined it to be incorrect (no), correct (yes), or unclear (unsure).

variant calls from the assembly. Since the currently-published (v3.3.2) short read-based GIAB benchmark almost entirely excludes the MHC[1], we compare the assembly-based diploid variant calls to the whole-genome mapping-based approach used to create the v4.1 variant benchmark set from GIAB that uses the same long reads and linked reads plus some additional short and long read data[27]. We find high concordance between the assembly-based method and the mapping-based methods over the regions accessible to short or long read mapping. Relative to the draft mapping-based benchmark, we report 7369 (49 %) more variants from the haplotig to reference alignments, so GIAB substituted an assembly-based variant benchmark in the MHC for the final v4.1 benchmark set. These additional variants are likely from those regions where HG002 has at least one haplotype that is highly diverged from the reference, making it challenging to map individual reads. These additional, challenging benchmark variants will help develop better algorithms to improve mapping and variant calling in these regions. Beyond the MHC, there are other challenging, highly variable regions like the KIR and IGH loci, as well as segmental duplications, that could benefit from future benchmark variant sets derived from a de novo assembly approach like the one demonstrated in this work.

As long read sequencing continues to become more accessible, a combination of long read and short read technologies for resolving difficult genomic regions in many individuals will become important. Robust genome characterization methods will help to investigate diseases that are still elusive when only considering simple variants[28]. With the recent Human Pangenome Reference Consortium to sequence 350 human genomes with long reads for de novo assemblies, our knowledge about the whole MHC region will increase rapidly. A pangenomics variant call benchmark for many individuals may become essential for economically genotyping the whole MHC region correctly. We hope the rich collections of diverse datasets and analyses for the GIAB samples and the future population-scale de novo sequencing will enable precision medicine from complicated genomic regions like MHC.

## Methods

**Cell line**. For the 10x Genomics and Oxford Nanopore sequencing, the following cell lines/DNA samples were obtained from the NIGMS Human Genetic Cell Repository at the Coriell Institute for Medical Research: GM24385. For the PacBio HiFi sequencing, NIST RM 8391 DNA was used, which was prepared from a large batch of GM24385.

**Recruiting WGS reads for the MHC region of HG002**. To identify WGS reads that belong to the MHC region, we selected the reads that are aligned to GRCh37 MHC regions without including alternative loci in the reference. Specifically, we retrieved reads from PacBio Sequel System HiFi 15 kb and ultralong Oxford Nanopore Technologies (ONT) datasets mapping to 6:28477797-33448354 and separated the reads based on the haplotype tag. It is possible that some MHC reads from HG002 are missed if they come from parts in the MHC region where HG002

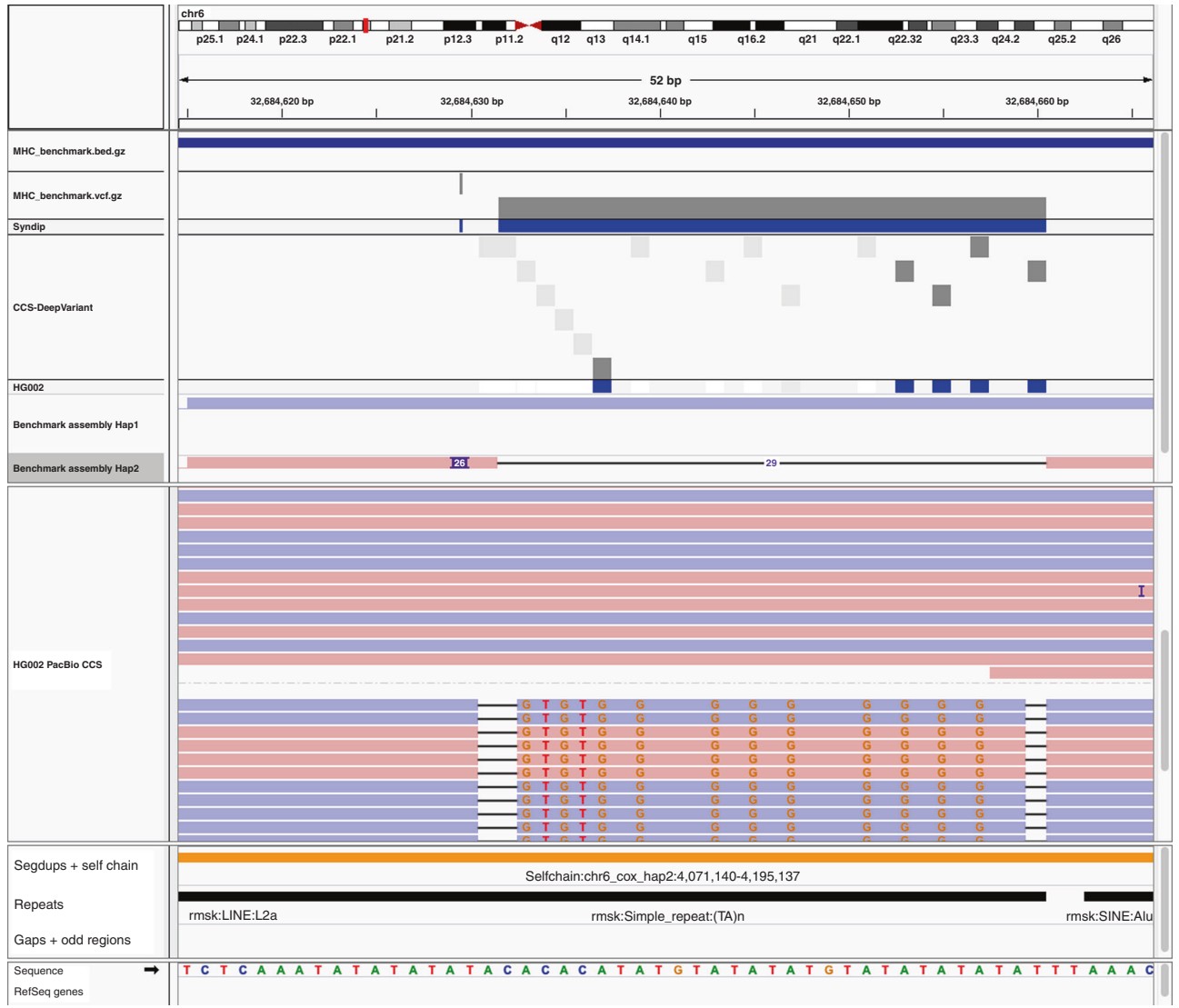

**Fig. 4 Example of partially called complex variant counted as both false positives and false negatives.** The CCS-DeepVariant VCF from PacBio HiFi reads incorrectly filters the 2-bp deletion and 9 of the 13 SNVs in the region (filtered variants are light gray boxes). The benchmark correctly calls this complex variant, and represents it as a 26-bp insertion of a TG tandem repeat followed by a 29-bp deletion of adjacent tandem repeats. When comparing this VCF to our MHC benchmark, the benchmark insertion and deletion variants are counted as false negatives, while the 5 SNVs called are counter-intuitively counted as false positives because the other variants are incorrectly filtered. If the CCS-DeepVariant VCF had not filtered all of the other variants, all variants would be counted as true positives.

is very different from the primary GRCh37 MHC region. In order to catch all possible HiFi reads that indeed belong to the MHC region of HG002, we also generated a de novo assembly of the HG002 MHC region using the Peregrine Assembler[29] and extracted reads that map to the de novo assembled contigs as unphased reads. (See Supplementary Note 3, 00_fetchreads.ipynb for the detail on recruiting the HiFi reads used for assembly and 01_get_phased_reads.ipynb for fetching ultralong ONT reads).

**Partitioning reads by haplotype**. We partitioned the reads associated to the MHC region by haplotype as follows. First, we established a set of high-confidence heterozygous SNVs by using two independent long read technologies and two different methodologies as shown in Fig. 1: We started from 12,846 bi-allelic heterozygous SNVs in the MHC region called by DeepVariant using HiFi data and regenotyped these SNVs using a haplotype-aware genotyping approach implemented as part of WhatsHap[30], run separately on both ONT and HiFi data. We retained 12,441 SNVs that were concordantly deemed heterozygous by these three approaches (DeepVariant with HiFi data, WhatsHap regenotyping with HiFi data, WhatsHap regenotyping with ONT data). To phase this set of high-confidence SNVs, we used WhatsHap to combine phase block information from 10x Genomics' LongRanger pipeline with ONT reads[23,24]. Next, we used WhatsHap *haplotag* to assign each HiFi and each ONT read to a haplotype with respect to this phasing. The initial fastq files for HiFi and ONT were then split by haplotype according to

this assignment (WhatsHap *split*), resulting in three read sets for each phased block (Haplotype 1, Haplotype 2, untagged). These haplotype-separated reads were subsequently used as input for the assembly process. We describe the Jupyter notebook of this workflow in Supplementary Note 3.

**Assembling haplotype-specific contigs**. We generate two haplotype-specific MHC assemblies for HG002 using the haplotype-partitioned reads and unphased reads. Namely, we generate each haplotype assembly from (1) reads with definite haplotype assignments, (2) reads aligned to GRCh37 MHC regions without enough variants for a definite haplotype assignment, and (3) reads recruited using de novo MHC contigs to catch sequences that may not be represented in the primary GRCh37 MHC region. We designate the two haplotype assemblies as H1 and H2. The H1 and H2 assemblies are generated from 7006 and 7043 haplotype-specific reads, respectively. Both assemblies share 1593 unphased reads recruited from alignments to the primary MHC sequence in GRCh37 and 335 unphased reads recruited from the HG002 de novo assembly. Due to (1) incomplete or erroneous segregation of the haplotype-specific reads and (2) recruitment of reads from other homologous loci (e.g., chr3: 143.15 Mb to 143.19 Mb and chr11:50.24 Mb to 50.28 Mb) to the MHC region, the assembly results usually contains smaller contigs (~30 kb) beside the major contigs (~5 Mb). We removed those spurious contigs for later analysis and created the benchmark callset with only the major contigs, one for each haplotype.

**Identifying low-quality regions by aligning reads to the haplotigs**. We can identify low confidence regions in the contig by checking the consistency between the reads and the assembled contigs. We align the reads back to the assembled contigs and call variants. In the ideal case, the assembled contigs and reads should be fully consistent and we should not observe any systematic differences (i.e., called variants) between the reads and the contigs. If there are problematic regions in the assembly, we might see clustered variants which are likely caused by incomplete partitioning or unresolved repeats in the assembly, as seen previously in haploid assemblies[6]. We use Minimap2[31] and pbmm2[18] to generate the alignments. We call SNVs from the alignment with FreeBayes v1.3.1-1-g5eb71a3-dirty and structural variants with pbsv. To identify problematic regions in the assembly, we filter variants to those with an allele fraction between 25 and 75%, cluster variants within 10,000 bp, and extend by 50 bp on each side. We find two regions >300 bp in size where the reads are inconsistent with each haplotig. The low confidence regions in the haplotigs and the corresponding regions in GRCh37 are listed in Supplementary Table 2.

**Making variant calls from the assembly**. To generate variant calls from the assembly, we used dipcall code from https://github.com/lh3/dipcall/releases version v0.1 with GitHub commit 7746f33. We modified the https://github.com/lh3/dipcall/blob/master/run-dipcall#L40 by adding -z200000,10000 (Supplementary Table 3) to increase mapping sensitivity of Minimap2[31] and calling of complex variants in the MHC region.

**Comparison to phased HLA typing**. As an independent evaluation of the phasing quality and base-level accuracy of the assembled H1 and H2 contigs for HG002, we used sequence-based classical HLA typing results for HG002, HG003, and HG004 generated by Stanford Blood Center[32] (Supplementary Table 4). Trio phasing of the HLA types was used to determine the paternal and maternal HLA haplotypes of HG002. HLA loci in the assembled haplotigs were identified and compared to classical HLA typing results with HLA*ASM, using minimap2 for the guide alignment step (parameter "–use_minimap2 1").

**Generating benchmark small variant set**. To generate benchmark small variant calls and regions for the MHC, we used the dipcall VCF and excluded 323,380 bp in several regions from the benchmark: (1) 37,157 bp in regions with variant calls identified when aligning HiFi reads from each haplotype back to the largest assembled contig from each haplotype; (2) 68,834 bp in regions with structural variants ≥50 bp in size, expanding to include any overlapping tandem repeats, plus 50 bp padding on each side, and merging regions within 1000 bp; (3) 18,574 bp in regions with >10 variants after clustering variants within 10 bp, and with >20 variants after clustering these regions within 1000 bp, plus 10 bp padding on each side, to exclude highly complex regions that might better be described as structural variants; (4) 87,318 bp in homopolymers, including imperfect homopolymers interrupted by single bases, longer than 10 bp in length plus 5 bp padding on each side, since these exhibit a higher error rate for HiFi reads; (5) 116,888 bp in the region containing the HLA-DRB genes because it had extreme structural divergence from the reference; (6) 119,086 bp in regions not covered by alignments of both haplotigs (outside the dip.bed output by dipcall). Finally, we exclude any homopolymers, tandem repeats, and low complexity regions that are only partially covered by the benchmark, which mitigates comparison problems related to errors in the benchmark for complex variants. To liftover calls from GRCh37 to GRCh38, we added 32,223 to the POS field of the VCF file and to the start and end positions in the bed file and removed the call at chr6:28719765, since the MHC sequence is identical between GRCh37 and GRCh38 except at chr6:28719765, which was changed from T to C in GRCh38.

**Evaluation callset: PacBio HiFi reads with GATK haplotype caller**. HG002 HiFi reads from three publicly available datasets (Table 1) were aligned to the GRCh37 and GRCh38 references using the pbmm2 v0.10.0 with '–preset CCS'. Small variants were called with GATK v4.0.10.1 HaplotypeCaller with '–pcr-indel-model AGGRESSIVE' and '–minimum-mapping-quality 10'. Variants were filtered on the QD (Quality by Depth) value with GATK v4.0.10.1 Variant Filtration, such that: (1) SNVs with QD < 2 are filtered, (2) Indels > 1 bp with QD < 2 are filtered, and (3) 1 bp Indels with QD < 5 are filtered.

**Evaluation callset: PacBio Hifi reads using minimap2 with DeepVariant**. A set of ~80x coverage PacBio CCS data was mapped to each reference:

    minimap2 VN:2.15-r905
    minimap2 -ax asm20 -t 32

(Note that the mapping of these files predates some improved recommendations for mapping to use pbmm2)

DeepVariant v0.8 with the PACBIO model was applied to the mapped files. The commands and workflow used are identical to the DeepVariant case-study:

https://github.com/google/deepvariant/blob/r0.8/docs/deepvariant-pacbio-model-case-study.md

No filtering is applied.

**Evaluation callset: PacBio HiFi reads realigned using Duplomap**. HG002 HiFi reads aligned to the GRCh37 reference using Minimap2 were downloaded from ftp://ftp-trace.ncbi.nih.gov/ReferenceSamples/giab/data/AshkenazimTrio/HG002_NA24385_son/PacBio_CCS_15kb_20kb_chemistry2/, and reads overlapping segmental duplications were realigned using a tool Duplomap (https://gitlab.com/tprodanov/duplomap) that used paralogous sequence variants to map reads with multiple possible alignment locations. Small variants were called from the realigned bam file using DeepVariant v.0.8 with default parameters.

**Evaluation callset: 10x genomics using Aquila local assembly**. Aquila uses linked-read data for generating a high quality diploid genome assembly, from which it then comprehensively detects and phases personal genetic variation. Here, Aquila merged two link-reads libraries to generate WGS variant calls for HG002/NA24385. Assemblies and VCFs for this merged library L5 + L6 can be found at http://mendel.stanford.edu/supplementarydata/zhou_aquila_2019/. The raw linked-reads fastq files can be downloaded in the Sequence Read Archive and its BioProject accession number is PRJNA527321.

**Evaluation callset: illumina TruSeq DNA PCR-free reads with illumina dragen Bio-IT platform**. Illumina PCR-Free reads (2 × 250 bp with 350 bp insert size) are downloaded from the public file server.

Dragen 3.3.7 is used to perform alignment, variant calling, and filtering on GRCh37 and GRCh38 reference assemblies. Variant filtering is based on MQ (Mapping Quality), MQRankSum (Z-score From Wilcoxon rank sum test of Alt vs Ref read MQs), and ReadPosRankSum (Z-score from Wilcoxon rank sum test of Alt vs Ref read position bias) values. For SNVs, MQ < 30.0, MQRankSum < −12.5, or ReadPosRankSum < −8.0 are filtered out. For INDEL, ReadPosRankSum < −20.0 are filtered. Illumina PCR-Free reads are downloaded from ftp://ftp-trace.ncbi.nlm.nih.gov/ReferenceSamples/giab/data/AshkenazimTrio/HG002_NA24385_son/NIST_Illumina_2×250bps/reads/.

**Evaluation callset: Illumina TruSeq DNA PCR-Free reads with VG alignment and Illumina Dragen Bio-IT platform**. Illumina PCR-Free read pairs (2 × 250bp with 350 bp insert size) are downloaded from and extracted from novoaligned bams that are hosted on the public file server. The process is based on aligning the HG002 to genome graphs that were constructed from HG003 and HG004 parental variants. All alignments are performed using Variation Graph Toolkit (VG) and variant calling is done using Dragen version 3.2. Default variant calling settings in

---

**Table 1 PacBio HiFi reads used in evaluation of benchmark.**

| Instrument | Insert Size | SRA | FTP |
|---|---|---|---|
| Sequel system | 10 kb | – | ftp://ftp.ncbi.nlm.nih.gov/ReferenceSamples/giab/data/AshkenazimTrio/HG002_NA24385_son/PacBio_CCS_10kb |
| Sequel system | 15 kb | SRX5327410 | ftp://ftp.ncbi.nlm.nih.gov/ReferenceSamples/giab/data/AshkenazimTrio/HG002_NA24385_son/PacBio_CCS_15kb |
| Sequel II system | 11 kb | SRX5527202 | ftp://ftp.ncbi.nlm.nih.gov/ReferenceSamples/giab/data/AshkenazimTrio/HG002_NA24385_son/PacBio_SequelII_CCS_11kb |

GRCh37 reference used for alignment: ftp://ftp-trace.ncbi.nih.gov/1000genomes/ftp/technical/reference/phase2_reference_assembly_sequence/hs37d5.fa.gz.
GRCh38 reference used for alignment: ftp://ftp.ncbi.nlm.nih.gov/genomes/all/GCA/000/001/405/GCA_000001405.15_GRCh38/seqs_for_alignment_pipelines.ucsc_ids/GCA_000001405.15_GRCh38_no_alt_analysis_set.fna.gz.
https://github.com/PacificBiosciences/pbmm2.
https://github.com/broadinstitute/gatk/releases/tag/4.0.10.1.

Dragen 3.2 were used during GVCF and VCF variant calling. The methods used to convert graph alignments to linear alignments and parental graph construction are in the workflow defined on the vg_wdl GitHub repository.

The workflow used to process this data can be found here https://github.com/vgteam/vg_wdl/blob/master/workflows/vg_trio_multi_map_call.wdl

Illumina PCR-Free reads for the trio used in parental graph construction and HG002 alignment are downloaded from

ftp://ftp-trace.ncbi.nlm.nih.gov/ReferenceSamples/giab/data/AshkenazimTrio/HG002_NA24385_son/NIST_Illumina_2x250bps/novoalign_bams/

ftp://ftp-trace.ncbi.nlm.nih.gov/ReferenceSamples/giab/data/AshkenazimTrio/HG003_NA24149_father/NIST_Illumina_2x250bps/novoalign_bams/ ftp://ftptrace.ncbi.nlm.nih.gov/ReferenceSamples/giab/data/AshkenazimTrio/HG004_NA24143_mother/NIST_Illumina_2x250bps/novoalign_bams/

The population data used for initial graph alignments of the HG002 trio samples are based on the 1000 genomes phase 3 variant dataset and the GRCh37 reference genome. http://ftp.1000genomes.ebi.ac.uk/vol1/ftp/release/20130502/ALL.wgs.phase3_shapeit2_mvncall_integrated_v5b.20130502.sites.vcf.gz

**Evaluation callset: 10x genomics using LongRanger with GATK haplotype caller**. These callsets, generated independently for each individual in the Ashkenazi trio, used LongRanger[21] (version 2.2, code at https://github.com/10XGenomics/longranger) and GATK v4.0.0.0 as variant caller with default parameters on 10x Genomics linked-reads data for the family trio (84x, 70x, and 69x coverage for HG002 NA24385 son, HG003 NA24149 father, and HG004 NA24143 mother, respectively) against both GRCh37 and GRCh38. The vcf and bam files for each genome are under:

ftp://ftp-trace.ncbi.nlm.nih.gov/ReferenceSamples/giab/data/AshkenazimTrio/analysis/10XGenomics_ChromiumGenome_LongRanger2.2_Supernova2.0.1_04122018/

The variant curation used the 10x Genomics VCF from LongRanger 2.2 derived from SRA accession SRX2225480 [https://www.ncbi.nlm.nih.gov/sra/SRX2225480], which is available at: ftp://ftp-trace.ncbi.nlm.nih.gov/ReferenceSamples/giab/data/AshkenazimTrio/analysis/10XGenomics_ChromiumGenome_LongRanger2.2_Supernova2.0.1_04122018/GRCh37/NA24385_300G/NA24385.GRCh37.phased_variants.vcf.gz

All samples were sequenced on the Illumina Xten at $2 \times 150$bp. The Ashkenazim trio was done using the v1 of the 10x library prep protocol.

**Evaluation callset: PacBio HiFi Clair**. This callset was generated using Sequel II 11kbp HiFi reads aligned to the hs37d5 reference with pbmm2, publicly available here: ftp://ftp-trace.ncbi.nlm.nih.gov/ReferenceSamples/giab/data/Ashkenazim-Trio/HG002_NA24385_son/PacBio_SequelII_CCS_11kb/. The variants were called by using Clair (v1) on these alignments.

**Reporting summary**. Further information on research design is available in the Nature Research Reporting Summary linked to this article.

## Data availability

All relevant data supporting the key findings of this study are available within the article and its Supplementary Information files or from the corresponding author upon reasonable request. The assembled haplotigs and benchmark variant calls and regions are available at: [https://github.com/NCBI-Hackathons/TheHumanPangenome/tree/master/MHC/]. While other PacBio data were used in the evaluation, described above, the assembly process used PacBio Sequel II System 15 kb (2 libraries) and 20 kb (2 libraries) CCS/HiFi data[18,32], which are available at: SRA accessions SRX7083056-SRX7083059 [https://www.ncbi.nlm.nih.gov/bioproject/PRJNA586863] and [ftp://ftp-trace.ncbi.nlm.nih.gov/ReferenceSamples/giab/data/AshkenazimTrio/HG002_NA24385_son/PacBio_CCS_15kb_20kb_chemistry2/]. The phasing process used ultralong ONT data, mostly from minion and gridion, which is available at: SRA accessions SRX7684921-SRX7685027 [https://www.ncbi.nlm.nih.gov/bioproject/PRJNA200694] and [ftp://ftp-trace.ncbi.nlm.nih.gov/ReferenceSamples/giab/data/AshkenazimTrio/HG002_NA24385_son/Ultralong_OxfordNanopore/guppy-V3.2.4_2020-01-22/]. The phasing process used the 10x Genomics VCF from LongRanger 2.2 derived from SRA accession SRX2225480, which is available at: [ftp://ftp-trace.ncbi.nlm.nih.gov/ReferenceSamples/giab/data/AshkenazimTrio/analysis/10XGenomics_ChromiumGenome_LongRanger2.2_Supernova2.0.1_04122018/GRCh37/NA24385_300G/NA24385.GRCh37.phased_variants.vcf.gz].

## Code availability

Jupyter notebooks are available for the analyses performed in this manuscript under: https://github.com/NCBI-Hackathons/TheHumanPangenome/tree/master/MHC/e2e_notebooks.

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

## Acknowledgements

We thank Ben Busby and Benedict Paten for leading the Pangenomics Hackathon at the University of California, Santa Cruz, where this work was initiated. We thank Heng Li for his recommendation to modify parameters in the Dipcall variant calling program. Certain commercial equipment, instruments, or materials are identified to specify adequately experimental conditions or reported results. Such identification neither does imply recommendation or endorsement by the National Institute of Standards and Technology, nor does it imply that the equipment, instruments, or materials identified are necessarily the best available for the purpose. Chunlin Xiao was supported by the Intramural Research Program of the National Library of Medicine, National Institutes of Health.

## Author contributions

C.S.C., J.W., Q.Z., E.G., S.G., M.R., M.S., T.M., A.T.D., and J.M.Z. designed the study. C.S.C., J.W., Q.Z., E.G., S.G., M.R., T.M., A.T.D., and J.M.Z. performed the analyses. C.X. managed the data. J.W., S.G., A.F., V.B., S.A., J.F., F.J.S., M.K., S.Z., A.C., W.J.R., M.C.S., C.M., B.Y., N.M., X.Z., A.M.B., and J.M.Z. performed the evaluations.

## Competing interests

C.-S.C. and A.F. are employees of DNAnexus Inc., a company providing a cloud computing platform for processing genomic information. C.-S.C. is a co-founder and partner of Omni BioComputing, LLC, which currently develops genome assembler related technologies. Q.Z. is an employee of Laboratory Corporation of America Holdings, a company providing clinical diagnostics services. A.T.D. is a partner in Peptide Groove, LLP. A.C. is an employee of Google, a company providing a cloud computing platform. W.J.R. is an employee and shareholder of Pacific Biosciences. A.M.B. is an ex-employee and shareholder of 10x Genomics. The other authors declare no competing interests.
