## [Peer Review File · Nature Communications]

Reviewers' comments:

Reviewer #1 (Remarks to the Author):

Dear colleagues,

I have read with interest your manuscript titled 'A Diploid Assembly-based Benchmark for Variants in the Major Histocompatibility Complex'.

The manuscript describes the construction of a reference dataset for use in benchmarking variation discovery approaches in the highly diverse and clinically relevant MHC locus in the reference HG002 individual. Using long read data from the two leading single molecule sequencing platform vendors (Pacific Biosciences and Oxford Nanopore) and linked reads (10x Genomics), the manuscript credentials a local de novo assembly approach to haplotig construction and the elucidation of variation contained within each haplotype. This new assembly-based benchmark is highly concordant with previous mapping-based benchmarks, but far more complete.

As mapping-only strategies are typically unable to access variation in loci greatly diverged from the reference, using a long-read assembly strategy to recover these missing 'personal' loci is eminently sensible. Further, the circular consensus sequencing (CCS) capability with the PacBio platforms now makes it possible to generate long reads with < 1% error rate. Coupled with ultra-long ONT reads and linked reads, your novel dataset and computational approach would appear to be highly useful for accurate assembly of this critical loci.

Comprehensive benchmark datasets are a critical tool for the development of new and improved variant calling methodologies, and as such, I believe these results are novel and will be of interest and utility to our community.

I also appreciate the extensive Github presence and Jupyter notebooks provided with the manuscript, which will be helpful for researchers attempting to reproduce this work.

There are a few points I would appreciate clarification on as they speak to the generalizability of the method to other samples, particularly those of diverse population backgrounds.

* In the Methods section, you state, 'To identify WGS reads that belong to the MHC region, we selected the reads that are aligned to GRCh37 MHC regions without including alternative loci in the reference.'. I'm curious as to why only the canonical locus is used and the alternative MHC loci are ignored. Is there an explicit reason for this? Would the consideration of the alt haplotypes significantly or only slightly increase the number of reads that presumably come from some real MHC haplotype?

* In the Results section, you state, 'In our experiment, we found that we needed to utilize all three data types [PacBio, Nanopore, 10x] to achieve a single phasing block through the whole MHC region.'. Can you speak to or demonstrate the relative utility of each successive addition of each of these technologies in establishing this benchmark dataset? Particularly as recent indications are that 10x Genomics will be discontinuing their linked reads for genomics kits, I am curious to know if MHC benchmarks made on different samples after this year will suffer greatly from the loss of the 10x component.

* In the Results section, you state, 'WhatsHap combined the long range information inherent to these data types to generate a single phased block for the whole MHC region without using parental sequencing from the trio'. Given that you have the data for the entire AJ trio, is there a reason it is preferable to not use it to phase reads in manner akin to how TrioCanu bins reads prior to assembly? Does the high heterozygosity in the MHC eliminate the need for the (presumably vastly more computationally expensive) trio binning approach?

* I note that previous long read assembly approaches that have achieved a high level of diploid MHC reconstruction (though likely not as complete as yours) include Jain et al. 2018 (<https://www.nature.com/articles/nbt.4060>) and Koren et al. 2018 (<https://www.nature.com/articles/nbt.4277>). (Note: I am not an author nor was I involved with either manuscript). Both of these papers present (among other things) analyses of MHC reconstruction in the GIAB sample NA12878. Neither of these manuscripts are cited by yours, despite seeming to be exceedingly relevant to the subject matter at hand. This brings to mind two questions:

1 - The first two sentences of the abstract state, ``We develop the first human genome benchmark derived from a diploid assembly for the openly-consented Genome in a Bottle/Personal Genome Project Ashkenazi son (HG002). As a proof-of-principle, we focus on a medically important, highly variable, 5 million base-pair region - the Major Histocompatibility Complex (MHC).'' Is this meant to mean, ``We have developed the first-ever MHC benchmark dataset using a diploid assembly' or, ``We have developed the first-ever MHC benchmark dataset using a diploid assembly *specifically for HG002*'? Should I not consider the MHC haplotigs presented in those manuscripts to be effectively MHC variant benchmarks for an openly-consented sample (with the missing step of distilling the haplotigs into a VCF)?

2 - How does your local assembly approach compare to those global assembly approaches? Is your procedure closing gaps in the MHC haplotigs that would remain in the ONT long+ultralong reads or PacBio-only assemblies presented by Jain et al. and Koren et al. respectively?

* Can you provide a supplementary table that describes the datasets in question, including read lengths, coverage, and error rates? In looking for some background information on the datasets in question, I found the statement ``As the CCS reads we used are around 13 kb in length', but a summary high-level description somewhere would be welcome.

Minor points:

* The sentence, ``We use each MHC haplotype to identify variants comparing assembly to assembly using pftools.' presumably is referring to pftools.js, not 'pftools'?

Reviewer #2 (Remarks to the Author):

Chin et al. develop a high-resolution diploid assembly based benchmark variant set for the MHC region of the HG002 Genome in a Bottle (GIAB) sample. Due to the complexity of MHC region in humans, it has been traditionally difficult to call variants or perform assembly in these regions and they are often excluded from analyses. To overcome these challenges, the authors utilize several unique sequencing data sets, including 10X Genomics Linked Reads, Oxford Nanopore long reads, and PacBio Circular Consensus reads, enabling the assembly of haplotypes covering nearly the entire MHC region, unlike the alternative contigs for the MHC from the GRCh38 assembly. The authors assess the local divergence of their assembled haplotypes from the MHC region of GRCh37 and call variants with respect to GRCh37 using Novograph. As part of efforts to identify high confidence "benchmark" regions for MHC small variants, these assembly based variants were compared to a non-assembly based draft GIAB variant call set from similar sequencing technologies, from which they attempt to identify clusters of false positive calls. The authors show that after restricting comparisons to high-confidence regions there are few differences between the alignment based and assembly based variant call sets. Finally, authors compare the assembly based benchmark calls to variants called using DeepVariant calls for the same sample derived from PacBio CCS reads, and using manual evaluation of a set of erroneous variant calls argue that the assembly based approach performs favorably.

In general, the abstract and discussion are well written, however the introduction lacks critical background information, and fails to put the work and its importance in the proper context for to be fully understood. Additionally, the results section is poorly written, and lacks critical interpretation and further analyses that could make this work of broader interest to the scientific community. It should be mentioned that the methods and code availability are excellent for this project. However, the manuscript could be improved by adding detail and clarity about the key datasets from Genome in a Bottle, and rationale for selection of computational methods (including how they compare to previous works). Furthermore, the utility of generating a phased assembly and benchmark variant call set should be described. The authors should also consider further interpretation of the benchmark variants and their functional impact, as well as description of structural variation identified within the MHC. Authors might consider refocusing the paper on their novel bioinformatics pipeline.

Introduction

The introduction focuses on the interesting biology and difficulty of variant detection within the MHC region (Paragraph 1), before describing how this benchmark dataset differs from previous mapping based approaches (Paragraph 2). Given that this manuscript is focused on focused on generating a diploid assembly-based benchmark perhaps the order of these two paragraphs should be switched and the emphasis modified. This is especially true because no novel biological insights about the MHC region are described in the manuscript.

The authors should describe the purpose and utility of benchmark variant call sets, perhaps describing some milestone benchmark variant call sets from GIAB. The authors should also remind the reader why short-read sequencing is insufficient for variant identification in "challenging" regions. Additionally, a brief overall of the sequencing methods used in this manuscript would be useful and how they are complementary for deriving diploid assemblies. The discussion is well written and contains some of this information – perhaps it can be moved.

Results

1. Linked reads and long reads generated a single phase block for the MHC

- The first paragraph of the results describes using several sequencing technologies to perform assembly, phasing, and variant identification. The authors should include some key details about these sequence data sets, including read length and depth add clarity to the following methods. Much of the information that should be in this paragraph is actually in the methods – Partitioning Reads by Haplotype.

- Figure 1 is not adequately referenced. The authors should reference specific panels of the Figure in the Results to help the reader understand the methods being described. The figure legend could be improved also. Again – some of the text in the methods – Partitioning Reads by Haplotype – could be put in the figure legend, which could be used as an overview of the study.

-The authors note that "we found that we needed to utilize all three data types to achieve a single phasing block through the whole MHC region." (Page 2). The authors should describe what was achievable with specific combinations of data types. As an aside – manuscripts should always be submitted with page numbers.

- The authors frequently reference approximate numbers in this section and others. For example "and about 20% of the MHC is covered by more than 10 of these reads due to runs of homozygosity and regions highly divergent from GRCh37. " (Page 2). The authors should provide precise numbers for these analyses.

2. CCS reads assembled into a single contig for each haplotype

- Author's note that there is a 30kb segmental duplication alignment gap containing several genes, noting that "It should be possible to manually incorporate ultralong ONT reads (which are long enough to traverse both copies of the repeat) to get the repetitive region assembled correctly, but

this will require future methods development. " (Page 4). Given the exclusive focus of this paper on the MHC, it would be appreciated if authors explored closing these alignment gaps, or described the challenge of doing so (in terms of methods).

- Authors describe second alignment gap of "a few hundred kb" (Page 4), again, authors should use specific numbers, and in this case provide coordinates of these highlighted alignment gaps.

- In Figure 2, authors should describe the calculation of estimated difference percentage (color scale).

- Authors should further describe how GRCH37 and GRCH38 differ at the MHC region. This could be done in the legend for Supplemental Table 1.

3. Assembled contigs completely match HLA types with correct phasing

- In Supplemental Table 3 the authors show the HLA genes called at 8-digit resolution. How did the authors call these genes at 8-digit resolution?

- Authors refer to "extra contigs" for the first time in this sentence "Since the main haplotigs (H1 and H2) matched the HLA types and covered the entire MHC region, and the extra contigs were short (close to the CCS read length), we disregarded these small contigs in further analyses.

" (Page 4). These should be referenced earlier when describing the assembly and Supplementary Table 1. How many extra contigs are there? How was it determined which haplotig they belonged to?

Create a reliable small variant benchmark set from the haplotigs

- A more detailed description of how the "draft v4.0" small variant benchmark was created would be appreciated. " we compared the Novograph assembly-based variant calls to a new draft v4.0 small variant benchmark set under development by GIAB, which uses mapped reads and variant calls from short, linked, and long reads. " (Page 5)

- As part of creating the small variant benchmark, and general variant calling, structural variants were identified. These were used to exclude regions from the benchmark but not described in the results. How many were there and where were they located? It seems like a missed opportunity to comment on structural variation within the MHC.

- Authors should provide a list of variants that were evaluated as FPs, FNs, or genotyping errors, and evidence used to determine that they fit these categories (at least for a subset of these cases).

- The authors describe an important issue when it comes to benchmarking- scoring complex variants. The extent to which this is an issue is not made clear however. How does hap.py vcfeval handle these situations? If it does not allow partial credit, would another tool be recommended?

"When benchmarking against dense variant calls in divergent regions like those in our MHC benchmark, it is critical to understand that current benchmarking tools will often classify a variant as a FP when both haplotypes are not fully called correctly in the query callset (e.g., if any nearby calls are filtered), since these complex variants can be represented in many ways and current tools will not always give partial credit if some parts of the complex variant are called correctly and some called incorrectly. " (Page 5)

Discussion

Typically, results are not given in the discussion for the first time, "we report 7668 (55 %) more variants from the haplotig to reference alignments." This is further indication that the results section need improvement.

Methods

The methods are well written, but some of the information in this section should be moved to the results section.

REVIEWERS' COMMENTS:

Reviewer #1 (Remarks to the Author):

Dear colleagues,

Thank you for your thorough responses. The new supplemental section on relative contributions from each technology are highly informative. I also appreciated the clarification on establishing a trio-independent MHC assembly procedure; I had not understood this goal before. The additional citations, expanded introduction, dataset summary, and other changes have also clarified the context and scope of the manuscript.

Your responses have appropriately addressed my concerns. I have no further comments.

- thank you for helping us improve the manuscript!

Reviewer #2 (Remarks to the Author):

The authors have addressed all comments/concerns. The manuscript is much improved.

Kelly A Frazer

- thank you for helping us improve the manuscript!

REVIEWERS' COMMENTS:

Reviewer #1 (Remarks to the Author):

Dear colleagues,

Thank you for your thorough responses. The new supplemental section on relative contributions from each technology are highly informative. I also appreciated the clarification on establishing a trio-independent MHC assembly procedure; I had not understood this goal before. The additional citations, expanded introduction, dataset summary, and other changes have also clarified the context and scope of the manuscript.

Your responses have appropriately addressed my concerns. I have no further comments.

Reviewer #2 (Remarks to the Author):

The authors have addressed all comments/concerns. The manuscript is much improved.

Kelly A Frazer

Thank you for your very helpful comments. Please see our responses below in red.

Reviewers' comments:

Reviewer #1 (Remarks to the Author):

Dear colleagues,

I have read with interest your manuscript titled 'A Diploid Assembly-based Benchmark for Variants in the Major Histocompatibility Complex'.

The manuscript describes the construction of a reference dataset for use in benchmarking variation discovery approaches in the highly diverse and clinically relevant MHC locus in the reference HG002 individual. Using long read data from the two leading single molecule sequencing platform vendors (Pacific Biosciences and Oxford Nanopore) and linked reads (10x Genomics), the manuscript credentials a local de novo assembly approach to haplotig construction and the elucidation of variation contained within each haplotype. This new assembly-based benchmark is highly concordant with previous mapping-based benchmarks, but far more complete.

As mapping-only strategies are typically unable to access variation in loci greatly diverged from the reference, using a long-read assembly strategy to recover these missing 'personal' loci is eminently sensible. Further, the circular consensus sequencing (CCS) capability with the PacBio platforms now makes it possible to generate long reads with < 1% error rate. Coupled with ultra-long ONT reads and linked reads, your novel dataset and computational approach would appear to be highly useful for accurate assembly of this critical loci.

Comprehensive benchmark datasets are a critical tool for the development of new and improved variant calling methodologies, and as such, I believe these results are novel and will be of interest and utility to our community.

I also appreciate the extensive Github presence and Jupyter notebooks provided with the manuscript, which will be helpful for researchers attempting to reproduce this work.

There are a few points I would appreciate clarification on as they speak to the generalizability of the method to other samples, particularly those of diverse population backgrounds.

* In the Methods section, you state, ``To identify WGS reads that belong to the MHC region, we selected the reads that are aligned to GRCh37 MHC regions without including alternative loci in the reference.'. I'm curious as to why only the canonical locus is used and the alternative MHC loci are ignored. Is there an explicit reason for this? Would the consideration of the alt haplotypes significantly or only slightly increase the number of reads that presumably come from some real MHC haplotype?

- GIAB uses the primary GRCh37 and GRCh38 assemblies without ALT loci because there are not yet standards for representing variants in or benchmarking against variants in ALT loci. We do expect that including ALT loci would recruit more reads, but the current MHC ALT loci are fragmented and could produce some analysis challenges (see figure below and blog post for more details at <https://medium.com/@infoecho/constructing-a-graph-for-genome-comparison-swiftly-d47dcd7eae5d>). We expect that the extra step of using the assembly to recruit more reads would still be needed for some individuals that have sequences highly divergent from the primary reference and the ALT loci, so we decided to use this strategy. We also anticipate there will be more complete MHC sequences finished in contrast to the relatively incomplete alternative contigs in GRCh38 in the coming years. We will be certainly interested in revisiting this in the future.

* In the Results section, you state, ``In our experiment, we found that we needed to utilize all three data types [PacBio, Nanopore, 10x] to achieve a single phasing block through the whole MHC region.' Can you speak to or demonstrate the relative utility of each successive addition of each of these technologies in establishing this benchmark dataset? Particularly as recent indications are that 10x Genomics will be discontinuing their linked reads for genomics kits, I am curious to know if MHC benchmarks made on different samples after this year will suffer greatly from the loss of the 10x component.

- Thank you for this suggestion. We have now added the analysis below as a supplementary note to justify the need for all 3 technologies:

To determine what technologies were necessary to obtain a single phasing block without using trio information, we used whatshap 0.18 with various combinations of technologies and calculated the number of phasing blocks:

- PacBio alone: Number of phased blocks: 30, Largest component contains 3095 variants (24.9% of accessible variants) between position 32553266 and 32910482

- PacBio + 10x: Number of phased blocks: 4, Largest component contains 6954 variants (55.9% of accessible variants) between position 28498559 and 31874134

- ONT ultralong alone: Number of phased blocks: 3, Largest component contains 7969 variants (64.1% of accessible variants) between position 28498559 and 32460863

- PacBio + ONT ultralong: Number of phased blocks: 3, Largest component contains 7969 variants (64.1% of accessible variants) between position 28498559 and 32460863 (Note, this results is the same as those of using ONT ultra long alone. We think PacBio reads do not contain extra phasing information on top of the ONT ultra long reads..)

- PacBio + ONT ultralong + 10x: Number of phased blocks: 1, Largest component contains 12441 variants (100.0% of accessible variants) between position 28498559 and 33448264

With our current methods and data types, we need all three data types to achieve a single phasing block across the MHC region.

The 10x data provides long-range information that could not be reached by PacBio + ONT so it is important for getting single phasing blocks. Our current focus is on using available data to get the improved benchmark set for HG002. We think OmniC or Hi-C data could provide similar information for phasing, although, eventually, we will still test them out empirically in the

following-up work. Meanwhile, new linked read methods such as BGI's stLFR technology could be also a replacement for 10x linked reads in the future.

* In the Results section, you state, ``WhatsHap combined the long range information inherent to these data types to generate a single phased block for the whole MHC region without using parental sequencing from the trio'. Given that you have the data for the entire AJ trio, is there a reason it is preferable to not use it to phase reads in manner akin to how TrioCanu bins reads prior to assembly? Does the high heterozygosity in the MHC eliminate the need for the (presumably vastly more computationally expensive) trio binning approach?

- We had initially explored using trio binning, but we decided to pursue the non-trio approach so that a similar benchmark could be created for any individual, even if parental data does not exist (e.g., for the GIAB HG003, HG004, HG006, and HG007 samples in the future). This approach also allowed us to use the parental data to help validate our phasing approach.

* I note that previous long read assembly approaches that have achieved a high level of diploid MHC reconstruction (though likely not as complete as yours) include Jain et al. 2018 (<https://www.nature.com/articles/nbt.4060>) and Koren et al. 2018 (<https://www.nature.com/articles/nbt.4277>). (Note: I am not an author nor was I involved with either manuscript). Both of these papers present (among other things) analyses of MHC reconstruction in the GIAB sample NA12878. Neither of these manuscripts are cited by yours, despite seeming to be exceedingly relevant to the subject matter at hand. This brings to mind two questions:

1 - The first two sentences of the abstract state, ``We develop the first human genome benchmark derived from a diploid assembly for the openly-consented Genome in a Bottle/Personal Genome Project Ashkenazi son (HG002). As a proof-of-principle, we focus on a medically important, highly variable, 5 million base-pair region - the Major Histocompatibility Complex (MHC).' Is this meant to mean, ``We have developed the first-ever MHC benchmark dataset using a diploid assembly' or, ``We have developed the first-ever MHC benchmark dataset using a diploid assembly *specifically for HG002*'? Should I not consider the MHC haplotigs presented in those manuscripts to be effectively MHC variant benchmarks for an openly-consented sample (with the missing step of distilling the haplotigs into a VCF)?

- We appreciate the reviewer pointing this out. We had intended to cite these manuscripts but neglected to do so in our submitted version. We've added a discussion of these papers as well as the 10x supernova paper to the introduction: "While human diploid assembly is currently making great strides, including fully resolving the MHC region in two haplotigs in two previous whole genome assemblies,(Jain et al. 2018; Koren et al. 2018) these still had a substantial error rates for small variants of at least 10 % due to their reliance on error-prone long reads, and the

individual long and ultralong read assembly incompletely resolved haplotypes. A linked read assembly also resolved much of the MHC for both haplotypes, but it was fragmented and had similar overall error rates for small variants(Weisenfeld et al. 2017). We expect this curated benchmark set from a targeted diploid assembly will help the community improve variant calling methods and whole genome de novo assembly methods, and form a basis for future diploid assembly-based benchmarks.”

2 - How does your local assembly approach compare to those global assembly approaches? Is your procedure closing gaps in the MHC haplotigs that would remain in the ONT long+ultralong reads or PacBio-only assemblies presented by Jain et al. and Koren et al. respectively?

- As described in the above added text, the published assemblies had relatively high error rates. That said, we expect whole genome diploid assemblies will continue their rapid improvement, and the targeted benchmark presented here will help optimize and demonstrate performance of these methods.

* Can you provide a supplementary table that describes the datasets in question, including read lengths, coverage, and error rates? In looking for some background information on the datasets in question, I found the statement ``As the CCS reads we used are around 13 kb in length', but a summary high-level description somewhere would be welcome.

We add, "We used the 10x Genomics Linked Read-based phased variant calls (84X coverage)²¹, Oxford Nanopore reads (ONT, 52X total coverage and 15X coverage by reads > 100kb)²², and PacBio Circular Consensus (HiFi, 18X coverage by 15kb library and 16X coverage by 20kb library) reads with predicted accuracy >99 %¹⁸ collected by GIAB (10x Genomics and PacBio) and UC Santa Cruz for establishing a high-confidence set of heterozygous marker SNVs, for phasing the corresponding variants, and generating haplotype-partitioned read sets with WhatsHap.¹⁷"

Minor points:

* The sentence, ``We use each MHC haplotype to identify variants comparing assembly to assembly using pfaatools.' presumably is referring to pafatools.js, not 'pfaatools'?

- Corrected to pafatools

Reviewer #2 (Remarks to the Author):

Chin et al. develop a high-resolution diploid assembly based benchmark variant set for the MHC region of the HG002 Genome in a Bottle (GIAB) sample. Due to the complexity of MHC region in humans, it has been traditionally difficult to call variants or perform assembly in these regions and they are often excluded from analyses. To overcome these challenges, the authors utilize several unique sequencing data sets, including 10X Genomics Linked Reads, Oxford Nanopore long reads, and PacBio Circular Consensus reads, enabling the assembly of haplotypes covering nearly the entire MHC region, unlike the alternative contigs for the MHC from the GRCh38 assembly. The authors assess the local divergence of their assembled haplotypes from the MHC region of GRCh37 and call variants with respect to GRCh37 using Novograph. As part of efforts to identify high confidence “benchmark” regions for MHC small variants, these assembly based variants were compared to a non-assembly based draft GIAB

variant call set from similar sequencing technologies, from which they attempt to identify clusters of false positive calls. The authors show that after restricting comparisons to high-confidence regions there are few differences between the alignment based and assembly based variant call sets. Finally, authors compare the assembly based benchmark calls to variants called using DeepVariant calls for the same sample derived from PacBio CCS reads, and using manual evaluation of a set of erroneous variant calls argue that the assembly based approach performs favorably.

In general, the abstract and discussion are well written, however the introduction lacks critical background information, and fails to put the work and its importance in the proper context for to be fully understood. Additionally, the results section is poorly written, and lacks critical interpretation and further analyses that could make this work of broader interest to the scientific community. It should be mentioned that the methods and code availability are excellent for this project. However, the manuscript could be improved by adding detail and clarity about the key datasets from Genome in a Bottle, and rationale for selection of computational methods (including how they compare to previous works). Furthermore, the utility of generating a phased assembly and benchmark variant call set should be described. The authors should also consider further interpretation of the benchmark variants and their functional impact, as well as description of structural variation identified

within the MHC. Authors might consider refocusing the paper on their novel bioinformatics pipeline.

Introduction

The introduction focuses on the interesting biology and difficulty of variant detection within the MHC region (Paragraph 1), before describing how this benchmark dataset differs from previous mapping based approaches (Paragraph 2). Given that this manuscript is focused on generating a diploid assembly-based benchmark perhaps the order of these two paragraphs should be switched and the emphasis modified. This is especially true because no novel biological insights about the MHC region are described in the manuscript.

- Thank you for this suggestion. We have reorganized these paragraphs as suggested, and have rewritten them to emphasize the rationale for the benchmark

The authors should describe the purpose and utility of benchmark variant call sets, perhaps describing some milestone benchmark variant call sets from GIAB. The authors should also remind the reader why short-read sequencing is insufficient for variant identification in “challenging” regions. Additionally, a brief overall of the sequencing methods used in this manuscript would be useful and how they are complementary for deriving diploid assemblies. The discussion is well written and contains some of this information – perhaps it can be moved.

- We have expanded the introduction to discuss the limitations of current benchmarks and short reads, as well as describe the complementary strengths of the linked and long read technologies used in this work.

Results

1. Linked reads and long reads generated a single phase block for the MHC

- The first paragraph of the results describes using several sequencing technologies to perform assembly, phasing, and variant identification. The authors should include some key details about these sequence data sets, including read length and depth add clarity to the following methods. Much of the information that should be in this paragraph is actually in the methods – Partitioning Reads by Haplotype.

- We have now added information about the sequencing datasets, and have moved some of the methods to the results as the reviewer correctly suggested.

- Figure 1 is not adequately referenced. The authors should reference specific panels of the Figure in the Results to help the reader understand the methods being described. The figure legend could be improved also. Again – some of the text in the methods – Partitioning Reads by Haplotype – could be put in the figure legend, which could be used as an overview of the study.

- We have revised the Figure 1 legend and referenced it more extensively in the results to help the reader follow the methods.

-The authors note that “we found that we needed to utilize all three data types to achieve a single phasing block through the whole MHC region.” (Page 2). The authors should describe what was achievable with specific combinations of data types. As an aside – manuscripts should always be submitted with page numbers.

- As included in our response to reviewer 1 above, we have now added a Supplementary Note 1 that gives the results for different combinations of data types. We have also added page numbers.

- The authors frequently reference approximate numbers in this section and others. For example “and about 20% of the MHC is covered by more than 10 of these reads due to runs of homozygosity and regions highly divergent from GRCh37.” (Page 2). The authors should provide precise numbers for these analyses.

- we have changed this and other approximate numbers to more precise numbers

2. CCS reads assembled into a single contig for each haplotype

- Author’s note that there is a 30kb segmental duplication alignment gap containing several genes, noting that “It should be possible to manually incorporate ultralong ONT reads (which are long enough to traverse both copies of the repeat) to get the repetitive region assembled correctly, but this will require future methods development.” (Page 4). Given the exclusive focus of this paper on the MHC, it would be appreciated if authors explored closing these alignment gaps, or described the challenge of doing so (in terms of methods).

- Thank you for this suggestion. We have continued to refine the assembly methods, and now we are able to resolve the this segmental duplication, so we now have no alignment gaps resulting from errors in the assembly: “We used the reads that were assigned to H1 or H2 and unphased reads as input for generating a haplotype-specific assembly. This resulted in two main haplotigs from two separate

assembly processes. Unlike most existing MHC alternate loci in GRCh38, these two haplotigs cover almost the entire MHC region (Supplementary Table 1). The alignments of the haplotigs to GRCh37 are shown in Figure 2. The alignments show a segmental duplication as well as several highly polymorphic regions, including a highly divergent region resulting in alignment gaps on both haplotypes..

There is a 30 kb segmental duplication in GRCh37 and both haplotigs containing the gene and pseudogene pairs RP, C4, CYP21, and TNX (RCCX). We use a two step assembly approach to resolve this highly similar segmental duplication. In the first step, we allow up to 4% difference between reads when building the read overlap graph. Due to the relatively large tolerance for differences, we can not distinguish the reads from different copies even though there are small differences between the copies. To resolve this, we introduce the second step for repeat resolution by analyzing the unique k-mers (k=32) of each read. We classify the k-mers to be (1) erroneous k-mers and (2) haplotype/repeat specific k-mers. With the repeat-specific k-mers, the reads from the two copies of the segmental duplication are separated before constructing the assembly graph."

- Authors describe second alignment gap of "a few hundred kb" (Page 4), again, authors should use specific numbers, and in this case provide coordinates of these highlighted alignment gaps.

Thanks for pointing this out. We report the alignment gap from the output of dipcall, as following in the revised manuscript:

"The alignments around the MHC Class I genes HLA-A, HLA-B, and HLA-C were very divergent, but <5% different so that the haplotigs were aligned without gaps. The only alignment gap occurred in the MHC class II genes in the 110 kb (H1), and 102kb(H2) between HLA-DRA and HLA-DRB1, caused by the extremely high divergence that frequently occurs in this region. "

- In Figure 2, authors should describe the calculation of estimated difference percentage (color scale).

We add a note on how we compute the differences in the figure caption as following:

"**Figure 2:** Alignments of the two main haplotigs to the primary GRCh37 MHC region. We compute the local divergence (est. difference) of the HG002 MHC haplotigs to the MHC of GRCh37 by performing local alignment. The differences between the assembled contigs and the references are computed using sequence blocks anchored with minimers and aligned locally using an O(ND) alignment algorithm."

- Authors should further describe how GRCh37 and GRCh38 differ at the MHC region. This could be done in the legend for Supplemental Table 1.

- We have added this note to the table: "The primary sequence of the MHC regions is identical in GRCh37 and GRCh38 (except at chr6:28719765 in GRCh38), but GRCh38 has additional ALT loci describing highly divergent sequences, which are in this table."

3. Assembled contigs completely match HLA types with correct phasing

- In Supplemental Table 3 the authors show the HLA genes called at 8-digit resolution. How did the authors call these genes at 8-digit resolution?

We are using the HLA-LA tool to identify the HLA type. We mentioned the command used to generate the call in the supplementary text. We add a reference in the caption for Supplemental Table 3 indicating how we make the HLA type calls.

HLA*ASM commands

```
perl HLA-ASM.pl --assembly_fasta $assembly.fa --sampleID $sample --  
truthFile truth.txt --use_minimap2 1
```

- Authors refer to “extra contigs” for the first time in this sentence “Since the main haplotigs (H1 and H2) matched the HLA types and covered the entire MHC region, and the extra contigs were short (close to the CCS read length), we disregarded these small contigs in further analyses. “ (Page 4). These should be referenced earlier when describing the assembly and Supplementary Table 1. How many extra contigs are there? How was it determined which haplotig they belonged to?

There 15 and 10 extra contigs for H1-asm and H2-asm respectively. In our evaluation we only take the longest contigs (~5Mb) in our downstream analysis. We have examined the possible causes for the extra contigs. The assembler is currently not designed to remove such contigs automatically. We add the following text in the methods to explain the cause and to provide the information about where the extract contigs mapped to:

“Due to (1) incomplete or erroneous segregation of the haplotype-specific reads and (2) recruitment of reads from other homologous loci (e.g. chr3: 143.15 Mb to 143.19 Mb and chr11:50.24 Mb to 50.28 Mb) to the MHC region, the assembly results usually contain smaller contigs (~30kb) in addition to the major contig (~5Mb). We removed those spurious contigs for later analysis and created the benchmark call set with only the major contigs, one for each haplotype.”

Create a reliable small variant benchmark set from the haplotigs

- A more detailed description of how the “draft v4.0” small variant benchmark was created would be appreciated. “ we compared the Novograph assembly-based variant calls to a new draft v4.0 small variant benchmark set under development by GIAB, which uses mapped reads and variant calls from short, linked, and long reads. “ (Page 5)

- we now compare to the release v4.1 benchmark set, and describe it in more detail. We are also writing a publication about this benchmark that will likely be in preprint form by the time this manuscript is published. If the reviewer is interested in an initial draft of this publication, it is at https://docs.google.com/document/d/12xTVZCftW2pPggtWr9QchiFABWzhW4-3A5sWR_5mt_s/edit?usp=sharing

- As part of creating the small variant benchmark, and general variant calling, structural variants were identified. These were used to exclude regions from the benchmark but not described in the results. How many were there and where were they located? It seems like a missed opportunity to comment on structural variation within the MHC.

- Thank you for this suggestion. We’ve added the following discussion of SVs as a Supplementary Note 2: “Although they are excluded from the benchmark bed, the dipcall vcf also includes 63 deletions and 63 insertions ≥50 bp in size. Upon curation of 20 randomly selected SVs, they all appeared to be accurate except for one assembler error in hap1, where the vcf has a false 55 bp deletion at 6:31690555 (near another false 27 bp insertion, both of which are excluded by the benchmark bed). However, 68 out of 126 are within 1000 bp of another SV, and 60 have at least 50 % overlap with a tandem repeat or homopolymer. Clustered SVs like these, particularly in tandem repeats, can typically be represented in many different ways, and unlike small variants, no benchmarking tools currently exist that correctly compare different representations of clusters of SVs. Therefore, we keep these in the vcf, but future work will be needed to develop tools to use these SVs to evaluate performance in an automated way. One complex example is an inversion and insertion in Supplementary Figure 1, which is represented by dipcall as a deletion at 6:31009222 and a compound heterozygous insertion at 6:31010095.”

- Authors should provide a list of variants that were evaluated as FPs, FNs, or genotyping errors, and evidence used to determine that they fit these categories (at least for a subset of these cases).

- We have added a much more extensive evaluation of the benchmark now, comparing 11 vcfs to the benchmark and curating FPs and FNs from each (see Figure 3). We have also added Supplementary Table 3 with the detailed curation results for each of the 210 curated variants.

- The authors describe an important issue when it comes to benchmarking- scoring complex variants. The extent to which this is an issue is not made clear however. How does hap.py vcfeval handle these situations? If it does not allow partial credit, would another tool be recommended? “When

benchmarking against dense variant calls in divergent regions like those in our MHC benchmark, it is critical to understand that current benchmarking tools will often classify a variant as a FP when both haplotypes are not fully called correctly in the query callset (e.g., if any nearby calls are filtered), since these complex variants can be represented in many ways and current tools will not always give partial credit if some parts of the complex variant are called correctly and some called incorrectly. “ (Page 5)

- We've added more details about this issue now, showing the number of partially correct calls in Fig 3B, and showing a more complex example in Fig. 4 that illustrates why no current benchmarking tools are able to give partial credit. We also now use dipcall instead of novograph for variant calling, because it represents complex variants as individual SNVs, insertions and deletions rather than as block substitutions. This makes it more likely that partial credit will be given for partially correct calls, but not in these most complex cases.

Discussion

Typically, results are not given in the discussion for the first time, “we report 7668 (55 %) more variants from the haplotig to reference alignments.” This is further indication that the results section need improvement.

- We've now added this result to the Results as well.

Methods

The methods are well written, but some of the information in this section should be moved to the results section.

- We have moved the results that were previously in the Methods into the Results section